# Exploring Community Perceptions of COVID-19 and Vaccine Hesitancy in Selected Cities of Ethiopia: A Qualitative Study

**DOI:** 10.3390/vaccines11101511

**Published:** 2023-09-22

**Authors:** Mulugeta Tamire, Teferi Abegaz, Samson Wakuma Abaya, Leuel Lisanwork, Lehageru Gizachew, Ebba Abate, Shu-Hua Wang, Wondwossen Gebreyes, Abera Kumie

**Affiliations:** 1Department of Preventive Medicine, College of Health Sciences, Addis Ababa University, Addis Ababa P.O. Box 9086, Ethiopia; teferiabegaz@gmail.com (T.A.); samson_wakuma@yahoo.com (S.W.A.); aberakumie2@yahoo.com (A.K.); 2Ohio State Global One Health, Addis Ababa P.O. Box 9086, Ethiopia; leuellisan79@gmail.com (L.L.); tadesse.43@osu.edu (L.G.); ebbaabate@yahoo.com (E.A.); 3Global One Health Initiative, The Ohio State University, Columbus, OH 43210, USA; shu-hua.wang@osumc.edu (S.-H.W.); gebreyes.1@osu.edu (W.G.); 4Infectious Disease Division, Internal Medicine Department, College of Medicine, The Ohio State University, Columbus, OH 43210, USA; 5Infectious Diseases Molecular Epidemiology Laboratory, Department of Veterinary Preventive Medicine, The Ohio State University, Columbus, OH 43210, USA

**Keywords:** COVID-19, community perception, vaccine hesitancy, Ethiopia

## Abstract

Even though the COVID-19 vaccine has been available and free of charge to the targeted population in Ethiopia, the vaccination rate was lower than needed to achieve herd immunity at community level. This study aimed to explore community perceptions of COVID-19 and vaccine hesitancy in selected cities of Ethiopia involving 70 in-depth interviews and 28 focused group discussions. The audio-taped data were transcribed verbatim, translated into English, and analyzed using a qualitative content analysis approach using the ATLAS.ti software version 8. The findings revealed that COVID-19 was perceived as evil and caused fear and frustration upon its emergence. The community initially used traditional remedies for its prevention but later transitioned to employing non-pharmaceutical interventions. The primary reasons for vaccine hesitancy were misinformation and misconceptions, such as connecting the vaccine with the mark of the beast, a lack of trust due to the multiple vaccine types, a shorter production timeline resulting in distrust of its effectiveness, and a fear of pain and side effects. Based on our findings, we recommend monitoring the use of social media and countering misinformation with the correct information and continuous public health campaigns. Further studies should be conducted to assess the types and magnitude of impacts from the myths and misconceptions on vaccination uptake.

## 1. Introduction

Since the emergence of severe acute respiratory syndrome coronavirus 2 (SARS-CoV-2), the Coronavirus Disease 2019 (COVID-19) pandemic has globally spread rapidly, causing increased death tolls and devastating social, economic, and political crises. It is well-acknowledged that non-pharmaceutical interventions (NPIs) have played an important role in flattening the epidemic curve, although the effects were not sustained, and the curve increased later in the pandemic [1]. In the absence of an effective treatment for any infectious disease, the best way to control the spread of the virus, lessen the socio-economic consequences, and mitigate the burden of the disease, is to use an effective vaccine [2]. However, at the time of this study the vaccination level in Ethiopia was lower than the required rate to achieve herd immunity at the community level to halt the spread of disease [3]. A very recent community-based study in five regions and Addis Ababa (the capital) also reported that below 30% of the total survey participants (1361) had been vaccinated with at least one dose of the vaccine [4]. 

The World Health Organization’s Strategic Advisory Group of Experts on Immunization has defined vaccine hesitancy as ‘a delay in acceptance or refusal of vaccination despite the availability of vaccination services’ [5]. Vaccine hesitancy is a rapidly increasing global public health challenge [6]. In many countries, vaccine hesitancy and misinformation pose substantial obstacles to vaccine coverage. Social media have increasingly become a source for spreading misinformation with certain groups exploiting negative attitudes towards vaccination and promoting conspiracy theories [7]. Misinformation has spread through multiple channels and social media platforms with considerable effect on the acceptance of a COVID-19 vaccine [6,8].

The degree of community involvement, social norms, perceptions of geopolitics, and vaccine nationalism also influence vaccine acceptance. The reasons for the low level of acceptance consist of religious, personal, or philosophical beliefs, safety concerns, and a need for additional education [9], with concerns about side effects being the most common reason for hesitancy [10]. Additionally, some are skeptical about the vaccine’s effectiveness and safety [11]; nearly 70% of vaccine hesitancy in Israel was due to safety problems [12]. Similarly, a low perceived risk of contracting COVID-19 linked to low mortality and severe illness rates as well as ideas about having natural immunity contributed to the refusal of the vaccine [13]. The public’s expectations are another issue for hesitancy if vaccination does not enable them to immediately discard their masks and return to life as it was in 2019. There was significant negativity towards the vaccination from the criticism of perceived coercive policies which can generate public frustration and anger [14].

To augment COVID-19 prevention in Ethiopia, a vaccination program was started in Addis Ababa. The first opportunity was given to healthcare professionals and the most susceptible population group in the community, including elders and those with chronic diseases. The COVID-19 vaccine was first introduced in Ethiopia on 13 March 2021, and delivered by health facilities [15]. Despite the offer of free vaccinations to the targeted populations at health facilities, achieving a first vaccination performance goal of at least 20% was not possible until November 2021. Vaccine hesitancy was highly characterized by limited trust in the vaccine’s efficacy and safety among healthcare workers. There was fear among the targeted population about receiving the vaccine due to rumors about side effects and effects on reproductive health and the possibility of other uncovered behaviors related to the knowledge, attitude, and practice of individuals [16]. Among medical students and health professionals, there was a 42% rate of vaccine hesitancy [17]. A recent study indicated a consistent low acceptance of the COVID-19 vaccine due to the worsening of existing health conditions and concerns that the vaccine itself could cause COVID-19 infections in Ethiopia [18]. 

After observing this hesitancy, the first campaign of 6.2 million doses began in November 2021 and was offered to the general population, including health professionals and vulnerable individuals aged twelve and above. The second campaign launched in March 2022, followed by a planned second and third campaign with a goal of a 60% vaccination coverage by the end of the 2022/23 year [15]. Four types of vaccines, namely, Sinopharm (Vero cell, Beijing, China), Oxford-AstraZeneca, manufactured by Serum Institute of India Pvt. Ltd. (Covishield, Hadapsar, India), Janssen Biotech Inc., a Janssen Pharmaceutical Company of Johnson & Johnson, Raritan, USA (Janssen), and Pfizer (BioNTech, New York, NY, USA), were approved for use in Ethiopia by the time of this study [19].

Available studies [16,17,18] concentrated on investigating vaccine hesitancy through a quantitative method wherein variables were utilized to define the level of knowledge and attitude towards the subject. These studies also focused on health professionals as the subjects for examining vaccine hesitancy. Given the considerable percentage of individuals who have low acceptance of the COVID-19 vaccine and the underwhelming results of the national vaccine campaigns, there is a need to explore in-depth the cultural beliefs associated with vaccine acceptance, which can be achieved through a qualitative approach.

Understanding community perceptions towards COVID-19 vaccination and the reasons for vaccine hesitancy would aid interventions aimed at increasing the vaccination rate to prevent a further spread of the disease. However, in Ethiopia—where vaccination rates remain low—neither community perceptions nor the reasons for vaccine hesitancy have been thoroughly explored. This qualitative study was conducted along with an ongoing quantitative study that assesses the trends in NPI-related community practices for the prevention of COVID-19 in respective cities in Ethiopia [20]. The purpose of this study is to investigate community perceptions and hesitancy towards COVID-19 vaccination in Ethiopia. The findings of the research will inform governmental and non-governmental organizations’ efforts towards COVID-19 prevention by providing reasons for future interventions and will serve as a baseline for future studies, both in Ethiopia and elsewhere.

## 2. Materials and Methods

### 2.1. Study Settings

At the time of this study, there were twelve regional states (currently thirteen) and two chartered cities in the Ethiopian government administration. This study was conducted in eleven major cities of Ethiopia, which were selected based on larger population size, high mobility, and availability of public transportation that favored the transmission of COVID-19. We included the two city administrations (Addis Ababa and Dire Dawa) and at least one city from every regional state of Ethiopia. In addition, these cities were part of our previous study to monitor the practice of non-pharmaceutical prevention for COVID-19 since the pandemic emerged in the country [20]. The distribution of these cities is indicated in Figure 1.

### 2.2. Study Approach and Period

We conducted a descriptive qualitative study using the qualitative content analysis approach [21] to explore perceptions and hesitancy towards COVID-19 vaccination. Since communities living in different regions and cities could have different socio-cultural backgrounds with different participant attributes, a qualitative content analysis is a more applicable method to analyze perceptions regarding COVID-19 and its vaccination. This study was conducted in August 2022.

### 2.3. Participants and Recruitment

In-depth interviews: We selected individuals for in-depth interviews from the general community or after visiting institutions selected for the quantitative NPI monitoring (observation), including banks, transport services, food and drink establishments, market centers, and religious institutions. We approached five people who had used the institutions and appointed them for an in-depth interview afterward.

Focus group discussions: Community members in cities were the participants for the Focus Group Discussions (FGDs), designed to create a platform for understanding collective thinking surrounding COVID-19 vaccination. The individual participants for FGDs were recruited based on their permanent residence in the area for at least six months and recognized as knowledgeable about their community regardless of gender and residence. Hence, research assistants recruited participants for in-depth interviews by themselves, using field facilitators or guides, who were residents and well placed to co-ordinate the selection of members of the community for FGDs. For the in-depth interviews, we included persons of different ages and genders while carrying out FGDs with at least one male and one female from each place.

Willing individuals with rich information were preferred to gain a deep understanding of the phenomenon under consideration. Purposive sampling techniques, and a mixed sampling technique with homogenous (by gender) sampling for FGDs and maximum variation (different age and gender) for in-depth interviews were employed. The selection of information-rich persons (intensity sampling) was also considered in both data collection methods. Experienced research assistants, who were university lecturers and had previous qualitative research experience, guided by field facilitators from each city, visited the settings and purposely chose individuals for the interviews.

### 2.4. Data Collection

A total of 70 in-depth interviews (IDIs) (with five participants from each city) and 28 FGDs (2 from each city, with an average of nine participants at each session) were conducted using the regional working language, namely, Amharic, Afan Oromo, Afar, and Somali. We included the participants from each type of institution used for the quantitative monitoring [20]. The 5 in-depth interviews and 2 FGDs from each city gave us the maximum data required to achieve saturation.

All data collectors had adequate qualitative research experience and previous experience conducting IDIs and FGDs; moreover, they all were university teachers holding a Master’s of Public Health degree or higher. Additional training and piloting of discussions were held prior to data collection to allow familiarization with the purpose of the study, interviews, and focus group guides.

The interview questions were mainly semi-structured, with some open-ended questions for further probing of responses. For the in-depth interviews and FGDs, the facilitator identified a place to ensure privacy and have a smooth discussion. All FGDs and interviews were conducted in noise-free places within the community and were audio-recorded. The seats for the FGDs were arranged in a circle to observe participant behavior and control group dynamics. Both the moderator and note-taker were seated in the circle facing each other to make eye contact, observe each other, and avoid status differences with participants during discussion. In situations where all were seated on the ground, our research team did the same, though two chairs were available. To be culturally sensitive, separate FGDs were conducted for males and females.

The moderators for the FGDs and in-depth interviews summarized the main discussion points at the end of each interview and FGD to the participants.

Notes or scribbles were taken during the in-depth interviews and later expanded to field notes. The field notes collected during FGDs were used to capture unspoken parts of the interview and other emotional reactions of the participants. Debriefing among data collectors and PIs, reviewing field notes, and listening to the audio were carried out daily to note preliminary findings and to identify areas to be further explored.

### 2.5. Data Analysis

After completing data collection, all audio recordings underwent verbatim transcription and were translated into English by experienced translators for analysis. To aid their understanding of the research’s scope and objectives, transcribers and translators were provided with a brief description of the data. The transcripts and translations were cross-checked against the audio recordings to ensure consistency. Familiarization started the analysis, before coding and analysis was carried out by experienced qualitative research assistants guided by the principal investigator with extensive qualitative research experience; the transcripts and field notes were read and re-read. Due to the large amount of qualitative data from numerous settings and groups, a codebook was prepared after reviewing the data and coding four transcripts. The final coding was conducted using an inductive qualitative content analysis approach with the ATLAS.ti version 8.0 software (Scientific Software Development GmbH, Berlin, Germany). We employed an inductive approach to establish sub-categories and main categories, which were derived from the data we collected. Finally, the study findings were compiled by adhering closely to the original context and meaning, and utilizing manifest meaning.

### 2.6. Trustworthiness

The trustworthiness of the data was ensured using different approaches, following Lincoln’s and Guba’s four criteria, namely, credibility, dependability, confirmability, and transferability [22]. To ensure credibility, data were collected from different cities and included individuals from various settings and backgrounds. The lead author, who has rich experience in qualitative research, oversaw the codebook development, data coding, and category development. Experienced qualitative research assistants who have completed Master’s degrees in Public Health and are teaching at the universities in each city, were also included.

To ensure the dependability of the findings, more than one individual was involved during both data collection and coding, and debriefing was conducted daily. The field notes, codebook, drafts of the findings, and the methodology were recorded to ensure confirmability and transferability.

### 2.7. Ethical Considerations

Ethical approval was obtained from the College of Health Sciences at Addis Ababa University, and permission to collect data was requested from the respective regional health bureaus. This was achieved by submitting a co-operation letter from the Federal Ministry of Health, which explained the aim of the study. Before commencing the study, all participants were asked to give their informed consent after being briefed on the objectives and importance of the research. They were also compensated with ETB 200 (USD 3.65) for their time and to cover phone communication (mobile card) and transportation costs. Lastly, the confidentiality and anonymity of all individuals involved in the study were maintained throughout.

## 3. Findings

This study involved 300 participants from 28 FGDs and 70 IDIs across ten regional cities and four sub-cities of Addis Ababa. The age range of the participants was 20 to 68. Of the participants, 160 (53%) were male and 140 (47%) were female. The majority had a primary school education background.

Six main categories and fourteen sub-categories (with three under the first two and eight under the third main category) were identified through analysis of the data (Table 1). The findings from the study were presented using the main categories accompanied by supporting quotes and organized accordingly. 

### 3.1. Early Perceptions of COVID-19 and Community Reactions

#### 3.1.1. Perceived as a Supernatural Punishment

The participants of the study stated that COVID-19 was initially perceived as an evil at the beginning of the pandemic. They mentioned that some religious leaders and social media influencers propagated views suggesting the pandemic was a supernatural punishment for sin, which the community accepted. It was also suggested that national prayers broadcasted on mass media and the state of emergency declared when the pandemic emerged in Ethiopia, also contributed to the acceptance of these views. They believed God intended to punish those who sinned and some even believed that Ethiopia would be protected from the disease and not affected by the evil.

*“When we heard that COVID had emerged and was present in our community, we became fearful and stressed. We were worried about how things might progress and feared for the health of our children. Generally, we felt stressed about how we were going to spend our time. Some of us even believed that it was God’s punishment for our sins.”* (Female FGD: AA P2)

There were participants who still believed the current time is bad and should pass with prayers. Most believed the burden of the disease was not as they had expected, compared with other developed countries, because of the power of prayer and the mercy of God. As a result, they continued to insist on praying rather than embracing other preventive measures.

*“This is an indication that people are not obeying God’s rules, and it is a sign of God’s anger towards us. Therefore, people must return to their prayers in addition to implementing all the prevention techniques.”* (Female FGD: DD P8)

#### 3.1.2. Experienced Fear and Frustrations

Many participants indicated they, and the community around them, were fearful and frustrated at the beginning of the pandemic. They mentioned that the community faced many tragedies. Some community members hid themselves upon observing symptoms of the disease, out of fear of isolation and related discrimination. Such events were reported by participants who observed their neighbors staying at home without informing anyone and ending up dying.

*“A man stayed hidden from the community in his home. Later he lost his appetite, had a cough, and felt general weakness. He was admitted to the hospital and eventually passed away.”* (Male FGD: BDR P6)

In Ethiopia, such events were observed during the first weeks of the pandemic. Health professionals transported suspected or confirmed cases to isolation centers by ambulance, resulting in no contact between patients and their families, as visiting the centers was not allowed at this time. This created a problem when the admitted patient did not have severe symptoms, but their family still could not visit them. Some thought the government was randomly taking people away, or that neighbors were purposely calling the ambulance when there was no sickness, and they were later convinced after watching others share their experiences of suffering in the media.

*“I felt frustrated when I heard about COVID-19; we even stopped working and stayed at home for several days.”* (Male FGD: BD P8)

#### 3.1.3. Higher Being: Feel Nothing

Some people were fearful without a valid reason, driven purely by emotion, saying nothing will come of it; some people related this situation to their own personal beliefs, stating that a supernatural force determines the date of death, and nothing can be done about it. Most of the discussants also stated that the incidence of disease and its burden, in terms of hospitalization and death occurrence, was not as severe as they initially anticipated. There were two different views reflecting this perception: one group mentioned supernatural protection, citing the national prayers of different religious denominations broadcasted on mass media; the other group linked it to the geographical location of the country, justifying similar cases happening in neighboring countries.

*“During the outbreak of COVID-19, we were all frightened. But everyone prayed according to their own religion. I would like to thank God for protecting me throughout the emergency.”* (Female FGD: AA P6)

### 3.2. Preventive Measures and Practice

The preventive measures implemented by the community were categorized into two groups: traditional and non-pharmaceutical measures. Additionally, there were a few participants who stated that protection was mainly provided by supernatural forces and individually they did nothing to prevent COVID-19.

#### 3.2.1. Traditional Methods: The Only Option in the Beginning

When the emergence of COVID-19 was declared in Ethiopia, the first community attempts, regardless of their place of residence, was to use some traditional remedies to protect themselves from the virus. The traditional remedies depended solely on some damakese (Ocimum Lamiifolium, common Ethiopian home remedy for headache, fever, pain, etc.), ginger, white onion, and herbal sources available in their living areas. According to some respondents, there were some social media sources that disseminated a list of traditional herbs and spices on the same evening of the declaration of the pandemic by the Ethiopian government, claiming the source was a known monastery, and that it was shared via digital media or through their neighborhoods. Others mentioned that the Prime Minister’s speech on that date, about using indigenous herbals and traditional spicy foods at household level for protection and reducing the effect of the disease, led the community to use traditional remedies. One of the participants listed all the traditional remedies as follows:

*“Our traditional spicy foods include data, karia, hariti, tena adame, yeketel bunna, damakese, ginger, and garlic. Ginger and garlic, in particular, are the ones we use the most in our daily activities. Generally, spicy foods and hot drinks help us strengthen our immunity for disease prevention.”* (IDI: HW)

Another source of traditional remedies was the use of Ethiopian indigenous fermented beverages, mainly areke and teje; some claimed areke, with its strong alcoholic content, protected them from acquiring the virus.

*“Our community also consumes alcohol, particularly areke, as it is potent and considered to lower the risk of acquiring the virus. We are aware that viruses, including the common cold, do not easily affect people who consume areke.”* (IDI: AS)

#### 3.2.2. Non-Pharmaceutical Interventions (NPI)

The participants listed various non-pharmaceutical measures practiced for COVID-19 prevention. Although we did not explore or request a demonstration of proper handwashing during the interviews or FGDs, the participants frequently mentioned washing their hands with soap or using hand sanitizers as their first response. In most cities, youth volunteers provided water and soap in public places and streets, but this was not sustained. Hand-washing facilities (water and soap) were also observed at the entrance gates of communal or some individual houses in the first weeks. Some indicated the complexity of the transmission sometimes made them feel ignorant. For them, washing hands whenever they touch surfaces was tiresome.

*“Using all the prevention methods was difficult during the COVID-19 emergency. There was a lack of hand sanitizers in the market, and if they were available, their prices would be high. Not only were sanitizer prices high, but also the prices of masks. For example, a mask could reach up to ETB 500 (USD 9.00). Sometimes it was impractical to wash hands after touching any surface, and it was difficult not to share latrines or water pipes.”* (Female FGD: AA P2)

Staying at home and maintaining physical distance were also mentioned as preventive measures, but some participants who relied on a daily income found it difficult to stay at home after a few days. Those with children at home were concerned about not exposing them, and some tried to change clothes in a separate room and wash their hands before having contact with them. Others went to such lengths as soaking a mat in bleach to wipe their shoes and even separating their beds within the home. One of the participants described the situation as follows:

*“At the beginning of the epidemic, I was very afraid. In fact, I didn’t even leave my house to reduce contact with others. However, over time, my fear of the disease diminished.”* (IDI: AW)

#### 3.2.3. Community Engagement on Preventive Measures

Various responses were obtained regarding community engagement in preventive measures, the level of which varied depending on the type of preventive measure. Based on these responses, all preventive measures were implemented in Addis Ababa. Participants from regional cities reported that hand washing and physical distancing were implemented at the beginning or first stage of the pandemic. At that time, community members panicked and tried to adhere to the preventive measures for a short period of time. However, engagement gradually decreased and almost no practice of preventive measures was observed in all regional cities and had nearly stopped in Addis Ababa during the collection of data for this study.

While respondents from some cities mentioned no or very low engagement in the use of face masks in their communities, others, mainly from Addis Ababa, reported full engagement due to the enforcement of the law requiring the use of face masks in public places and institutions to receive services; however, the level of engagement decreased after a few months. The respondents mentioned that the initial campaign included various strategies such as utilizing mass media and other social media to encourage community-level utilization.

*“In the beginning, we were washing our hands, wearing face masks, avoiding public places, and keeping social distance. We would even run away when a person sneezed. However, later, we stopped doing all of these things.”* (IDI: JG)

Some terminated the use of preventive measures, thinking that the impact of the disease was not as expected, while others said that there was no COVID-19 in Ethiopia. Most of the campaigns and voluntary efforts by the youth or other community members did not sustain, given what was considered an absence of the disease in the country. Some participants were saying that the pandemic hit the country like other neighboring countries, though not as strongly as Western nations.

*“I personally believe that the impact of the disease in Ethiopia will be significant, given its devastating consequences in Italy and the USA. As we know, their lifestyles are much better than ours. However, thanks to God, it has not caused major damage so far, even though it has not been completely eradicated yet.”* (Female FGD: ADK P2)

Now, there is no concern or commitment to use any source of preventive measures, and participants believe that neither healthcare providers nor the government expects them to practice it. In most cities, people may get confused if anyone uses a face mask or tries to maintain physical distance, but others have stated that they are still not comfortable when greeting strangers by hugging, and sometimes they do not even try it with anyone. They believe the culture of greeting other people has changed after the emergence of COVID-19. Predominantly two ideas were mentioned regarding the lack of sustainability of practices in the community: on the one hand, loose law enforcement and media campaigns regarding the practices were mentioned, while on the other hand, the community assumes that there is no COVID at all.

*“The reason for not giving attention to the disease is that government activities related to it have stopped. There is no news on TV or radio, unlike before.”* (Female FGD: HS P2)

### 3.3. Vaccine Awareness and Sources of Information

#### 3.3.1. Vaccine Awareness

Participants were asked about their knowledge of COVID-19 vaccination and the sources from which they obtained the information. All participants had awareness that a vaccine for COVID-19 existed and knew where it was offered. Some participants were uncertain about whether the vaccine was free of charge or not, as well as if there was a service fee. Others mentioned that some residents in their village failed to acquire the vaccine during their visit to a health facility. Later, some participants stated that this was due to the initial eligibility criteria, which covered only health workers and elderly individuals.

*“Initially, we heard that the vaccine would only be given to health professionals. Later, authorities tried to disseminate the vaccine to the community based on their priority. In addition to health professionals, the elderly and those with chronic diseases were given priority to be vaccinated first.”* (IDI: DD)

#### 3.3.2. Sources of Information about the Vaccine

Various sources of information for COVID-19 vaccination were mentioned, including media, health professionals, volunteering group campaigns, family, and friends. Mass media was found to be the primary source of information, followed by health professionals.

*“Every community has heard about the COVID-19 vaccines through mass media and television. In addition, health extension workers have also been informing us.”* (Male FGD: AS P2)

Although our study mainly targeted and involved non-vaccinated community members, the participants indicated that vaccine uptake and vaccination practices are still low at the community level. Some participants mentioned that they do not have any community members or neighbors who are vaccinated for COVID-19, while others said that some of their family members are vaccinated.

*“The number of individuals who have not been immunized is greater. In my family of nine, only two have received vaccinations. Therefore, the remaining seven have not been vaccinated. Similar situations exist within the larger community.”* (Female FGD: AA P1)

With regard to their attitude towards the benefits of the vaccine, some participants mentioned that getting vaccinated is helpful even though it does not completely prevent contracting the disease. One of them expressed the view that the vaccine could lower the intensity of the illness and individuals who are vaccinated might have higher chances of surviving.

*“There is a significant discrepancy between vaccinated and unvaccinated individuals. Vaccinated individuals can easily recover from viruses if they become infected, while unvaccinated individuals tend to experience worsening symptoms after getting infected. In some cases, this can even lead to death.”* (IDI: AS)

### 3.4. Reasons for Vaccine Hesitancy

#### 3.4.1. Misinformation and Misconception

The main reason for hesitancy towards COVID-19 vaccination is linked to the diffusion of misinformation and misconceptions among the community. Specifically, followers of the Christian religion, regardless of denomination, have expressed concerns that the vaccine may be used to administer the mark of the beast (666) on believers without their knowledge and make them followers of the Illuminati. Participants reported hearing this information from religious leaders, digital media, and community members.

*“We have heard that the vaccine is the devil’s work and its idea stems from 666. In my view, both the disease and the vaccine can be attributed to 666. We are concerned that the vaccine may cause serious complications in our lives in the future.”* (Male FGD: HS P4)

Others speculated that the vaccine has microchips, and the aim is to control the movement of people in low-income countries rather than prevent the disease. Some participants have heard this information from social media and some mainstream media, despite not knowing what a microchip is.

*“I believe God will protect me. I’ve heard that there are microchips being used to control our movements and that the disease is no longer a problem or is already being managed. Some people think that the vaccine itself is giving us the disease.”* (IDI: AA)

Some participants also held the belief that the vaccine could cause infertility in females; the misconception is that Western countries have a hidden agenda of population control in African nations.

*“This COVID-19 vaccine causes infertility in female. It makes women sterile (do not give any birth). Due to this condition, people do not receive COVID-19 vaccines.”* (Female FGD: AS P8)

A participant from Bahidar mentioned that such a suspicion has been in place for some years now and was more boldly stated during COVID-19.

*“I also believed that developed countries used this to minimize the population of developing countries. So that they create the disease deliberately.”* (IDI: AA)

The negative information related to COVID-19 vaccination was not limited to the above-mentioned misinformation. In some instances, participants heard that vaccinated individuals would die within two years, which led them to opt for risking the disease instead of getting vaccinated. During an in-depth interview, a participant from Addis Ababa explained this rumor:

*“I was not vaccinated because I heard rumors that said after two years of vaccination, it could result in death.”* (Male FGD: HR P3)

#### 3.4.2. Lack of Trust

Participants, particularly those educated with a health professional background, expressed a lack of trust in being vaccinated. Some mentioned the presence of too many different types of vaccines, while others cited concerns about the testing timeframe being insufficient for a new vaccine, especially after experimenting on various animals. The former group also mentioned that some countries were even banning the use of vaccines from certain companies, while others merely criticized them. The availability of these debates and criticisms on mass media and social media platforms caused confusion and further eroded trust in getting vaccinated. A participant from Addis expressed this sentiment:

*“Most of the community has not taken the vaccine. One of the reasons given is that they do not believe in the efficacy of the vaccine, which is related to the community’s spiritual beliefs. There are also concerns that the vaccine may cause infertility in women. Furthermore, the availability of different types of vaccines from various countries has caused confusion within the community, making it difficult for them to choose the best option.”* (Female FGD: HS P7)

The latter group mentioned that the time it took to develop the vaccines and make them available in the market for vaccination was very short. They suspected that the experimentation was not properly conducted before distribution. To justify this, participants indicated that researchers from France suggested in the media that testing the vaccines in Africa was a possibility. As a result, they suspected that they were to be used as subjects for the experiment or that some vaccines were available for donation while others were not. This led to their conclusion that a vaccine not thoroughly evaluated was sent to Africa.

*“I heard on social media that some researchers from France suggested testing new vaccines or drugs in Africa. How can I trust them after hearing this?”* (IDI: AS)

Therefore, some of the participants waited for a while to observe others getting vaccinated and to see if any problems arose. Later, they were vaccinated after it had been proven to be safe, whereas others were not vaccinated due to some of those who had already been vaccinated over-exaggerating their pain (such as joint pain) and staying at home off work. Additionally, there were groups who did not trust the vaccine or believed it was of poor quality, as it was available for free through donations. These individuals complained that the vaccine was not effective for them, or that it might have other problems leading to unknown health issues in the future. Lastly, some participants confidently expressed their views that the vaccine was made to kill Africans and they did not believe it was effective.

*“I specifically don’t agree with the effectiveness of the vaccine because a community member got sick and died after taking it. Not only me, but most of my neighbors also agree with this. We all believe that the Westerners created this vaccine to kill Black people, especially Africans. So, I didn’t take the vaccine because I believe traditional prevention methods will protect me rather than taking a vaccine.”* (Male FGD: HS P5)

#### 3.4.3. Afraid of Pain and Side Effects

Exaggerating the pain at the injection site and complaining about joint pain have been mentioned as reasons for vaccine hesitancy. In a few interviews and focus group discussions, the mode of vaccine administration and a fear or phobia of needle injections were cited as reasons for not getting vaccinated. Those who fear injections mentioned that they are even afraid when they need injections for other illnesses; some also stated that they have allergic reactions to injections and prefer to take medication in tablet form.

*“The pain experienced after vaccination is another factor that has been raised by individuals. There is also a rumor that if someone is already infected with COVID-19 and receives the vaccine, it could result in death. Due to these factors, many individuals choose not to receive the vaccination.”* (Female FGD: GA P7)

Others associated the mode of administration with the pain in their arms after taking the vaccine indicating that either their neighbors or other family members complained of the pain in their arms for a few days, making them reluctant to receive the vaccine. However, other FGD participants disagreed and argued that individual decisions were lacking, instead saying that localized pain after any type of injection might occur and was not an issue for them.

*“In the community, we hear different kinds of information about the side effects of vaccinations. They are said to cause dizziness, seizures, blood clotting, half paralysis, and a feeling of sleepiness. One of my friends informed me that his actual situation was due to the side effects of the vaccination.”* (Female FGD: AW P6)

To this end, some even associated pain felt weeks later with the vaccine while it could have other causes or reasons.

*“I did not get vaccinated; however, my mother received the vaccine and did not experience any symptoms immediately after. However, after a week, she experienced pain in her leg, which she had before. At this point, our neighbors suggested that it could be due to the vaccine. My mother became very afraid of the rumor that she could be paralyzed.”* (Female FGD: DD P6)

#### 3.4.4. Political Bias

Some members of the community did not believe in the occurrence of the pandemic and the need for vaccination, citing two reasons. As the pandemic emerged in Ethiopia a few months before the national election, some community members claimed that the government was announcing the emergence of the disease to avoid the election as there were no cases reported in the country. Study participants indicated that this view was orchestrated by certain YouTubers and prominent social media agents. Another group claimed that a supernatural force was punishing the government and putting pressure on the new leadership. However, most participants stated that this perception was wrong at the beginning of the pandemic, and they now believe COVID is real. The number of deaths in the country and the higher rate in other parts of the world are proof of this reality.

*“I don’t know anyone who is infected with COVID-19. Therefore, I believe it’s more of a political game rather than an actual existential threat.”* (Female FGD: GS P7)

#### 3.4.5. Lack of the Right Information for Decision: Pregnancy and Lactating Women

Participants who were pregnant or lactating at the time of the vaccine launch and after the entire community became eligible, mentioned that they did not receive the correct information to decide their eligibility for vaccination. They stated that health professionals did not provide them with clear information regarding their eligibility when they sought advice at health centers or hospitals. Some health professionals advised them to make the decision themselves, so the mothers chose to avoid it as the professionals failed to provide the correct information; some even advised against vaccination.

*“I went to the health facility to receive the vaccination, and prior to that, I inquired if lactating women were eligible for the vaccination. The healthcare professionals proceeded to discuss the matter in private with each other, and I became frightened and left without getting vaccinated. The healthcare professionals didn’t provide me with a confident response regarding the matter.”* (Female FGD: AA P2)

Others also noted no information regarding the vaccine in the media or even at health facilities. One participant in an in-depth interview explained the situation:

*“The vaccine was in campaign once but later on; we don’t have any clear information where this vaccine is given. We don’t have clear information on which type of the community gets the vaccine, where it is given and even there should be clear information whether it is given or not. Especially currently, we don’t have any clear information on vaccine accessibility.”* (IDI: AK)

#### 3.4.6. Economic Reasons

Some participants, mainly those who made their sources of income on daily work by themselves or being employed, indicated they were not vaccinated because they did not want to be absent from the job. The economic reason for non-vaccination is related to the misinformation about the pain and side effects of the vaccination resulting in an inability to work. There was common misinformation—the vaccine included the virus itself, leading to disease occurrence and inability to work; missing work for a few days due to the pain after vaccination. During the discussion, there were participants who still do not want to be vaccinated as they saw some individuals who were off work due to vaccination. One of the male FGD participants from Assosa, a masonry worker in the city, stated:

*“Everybody should survive with their daily labor, so those people receive a COVID-19 vaccine and become ill. So, how can they survive and serve their household members? I am a masonry worker, and I must work every day and I was not vaccinated in fear of that.”* (Male FGD: AS P7)

#### 3.4.7. Misperception of Absence of Symptom among Positive Cases

Related to this was the absence of symptoms among some patients where some family members were free of symptoms; thus there was a hasty generalization that the disease does not affect the younger age groups. However, there were participants who witnessed the occurrence of symptoms among the youth and their hospitalization or even death in some cities.

One of the IDI participants from Yeka, Addis Ababa, confidently said that COVID-19 is [like] a common cold, and he can easily resist the disease without vaccination.

*“My opinion, as it is not necessary to take the vaccine. So, I didn’t take the vaccine and I believe that I can resist the virus or the disease naturally like other diseases. For example, the common cold. Therefore, I think COVID-19 virus is also the same as other diseases for me, so I can resist [it]. My attitude towards the vaccine is like this.”* (IDI: AK)

#### 3.4.8. Lack of Awareness and Negligence

There were participants who said that the community had inadequate awareness about the vaccine and its advantages and disadvantages.

*“… Most part of the community didn’t take the vaccine, and this is due to the community’s poor awareness and acceptance regarding the virus and the disease.”* (IDI: DD)

Another female FGD participant from Semera, Afar, stated that she had no awareness about the vaccine as follows:

*“I don’t know about the COVID-19 vaccine and didn’t take the vaccine. The reason why I’m not utilizing the vaccine is that I have no information about the vaccine availability”* (Female FGD: SM P5)

Some of the participants said that their own negligence is the reason for not being vaccinated against the disease. One participant from Addis Ababa strictly mentioned that negligence is the main reason for his status of being not vaccinated.

*“I am not vaccinated because of my negligence; no other reason. I know the vaccine is available freely at the health centers.”* (IDI: AA)

### 3.5. Future Intentions towards Vaccination

The participants’ intentions to be vaccinated in the future reflected two opposing views. Those who agreed to take the vaccine in the future had adequate education and saw others not facing any problems. For this group, the reason was mainly the negative message and lack of counter-message to boost their confidence in agreeing to be vaccinated if the disease is sustaining for long in the future, but they had not yet decided clearly.

However, others insisted on their stand of not being vaccinated at all, justifying it either through a belief in supernatural forces protecting them from any disease or a lack of trust in the vaccine whatever justification is given for it. There were participants who associated the risk of disease and death with the will of supernatural forces and believed that whether someone is vaccinated or not the will of the Almighty is not changed. One of the FGD discussants in Assosa stated:

*“Our life is in the hand of God or Alaha, no one can interfere with it. If the health workers advise us for time, being we said yes, but after that we do not go to the vaccination service.”* (Male FGD: AS P1)

Another FGD participant from Addis Ababa added that using holy water is enough to protect oneself from COVID; she said that she would rely only on using it rather than vaccination.

*“I believe in holy water, so most of the time I use it, I don’t have the fear of being infected with COVID-19, but if it happens, I think I will recover by using holy water. So far, I haven’t been infected with COVID-19 so far.”* (Female FGD: AA P5)

However, some participants complained that relying on a supernatural force and not following preventive behavior or not being vaccinated has no religious basis. Their reason was that protecting oneself from diseases or any other bad things is a personal responsibility. One Muslim participant from a male FGD in Bahirdar mentioned a story about an occurrence of an outbreak when the Prophet Mohammed isolated patients from those who were healthy.

*“I am a Muslim. In the Quran [Koran], there was an outbreak and Prophet Mohammed isolated patients from healthy ones.”* (Male FGD: BD P8)

There were also others who believed that COVID-19 is over and therefore there was no need to be vaccinated. Those with this type of belief mentioned that the media and law enforcement activities have stopped because the disease has disappeared. One of the IDI participants in Addis Ababa stated:

*“Nowadays, the COVID situation is over, so I don’t think I will take the vaccine.”* (IDI: AK)

### 3.6. Suggestions on the Way Forward

We asked the participants to give their own suggestions on the way forward to address COVID-19 vaccination and other similar issues. Various recommendations were made for increasing the uptake of the vaccine including working with religious institutions and schools, channeling the correct messages about side effects through the media, continuing to create awareness, training the healthcare providers to give correct information, conducting more research on traditional medicines (believing that they can prevent COVID-19), and strengthening community engagement in the woreda.

A female discussant from a FGD in Addis Ababa mentioned the following on the need to create awareness and counter the rumors:

*“The health facilities must give briefing for the community on the COVID vaccine. They should be aware of us and give explanation on all the rumors that we hear. In addition to the health facilities, the media should work strongly like before.”* (Female FGD: AA AK P1)

Another participant also stressed the importance of proper information provision by the health providers in the localities as follows:

*“The health professionals should give appropriate information for the community.”* (Male FGD: HS P10)

## 4. Discussion

This study aimed to explore community perceptions of SARS-CoV-2 and hesitancy towards COVID-19 vaccination in the main cities of Ethiopia. Studies conducted during and after the COVID-19 pandemic indicate that a lack of social connectedness and loneliness had the strongest association with mental health during the pandemic [23,24,25]. The study participants had similar reflections and recognized that tragedies were happening at the community level, along with the social and economic impacts of COVID-19 at the beginning of the pandemic. There were community members who fled to rural areas from urban settings, with some hiding at home or becoming hopeless as they could not work outside for their daily income. Such encounters also affected social connectedness and other cultural values, including the local use of communal latrines and drinking water points.

All participants mentioned that their communities, as well as themselves, were in a state of panic at the beginning of the pandemic. There were tendencies to engage in non-pharmaceutical prevention methods, especially hand washing, physical distancing, and staying at home during that time. However, that adherence to public health recommendations decreased over time based on our previous trend analysis [20], and our observations during the period of this study also showed a decrease in the practice of hand washing, physical distancing, and mask-wearing in the regional cities and very limited practices in Addis Ababa.

In this study, the use of unproven traditional remedies from herbal sources and the use of Ethiopian indigenous traditional fermented beverages, especially areke, were mentioned across all settings. Such practices without adequate supporting scientific evidence might have contributed to the increase in occurrences of the disease and its related impacts. In addition to practicing preventive measures, holding such beliefs can affect the uptake of vaccination in the future. A literature review also reported the potential use or impact of natural products, as self-medication requires extensive research followed by clinical studies in collaboration with traditional medical practitioners [26].

The lack of maintenance in preventive practices and community engagement may be linked to the withdrawal or loosening of law enforcement at the national level, as well as the absence of continuity in health communication and community sensitization, including within the mass media. Additionally, the wrong interpretation of early interventions, such as travel bans, school closures, and other social, cultural, and religious service shutdowns aimed at preventing the spread of the disease, can lead to panic in the community. The perception of a greater burden than had occurred can also result in less adherence to preventive measures. Other studies have also reported a lack of sustainability in implementing COVID-19 preventive measures in Ethiopia and China [27,28].

The views regarding supernatural protection and a lower severity of the disease burden may influence future interventions to address similar emergencies. Therefore, the Ministry of Health and other researchers need to address these beliefs to ensure effective prevention measures are put in place. The community’s engagement in vaccination against the disease may be affected by such beliefs, indicating the need for religious leaders to be actively involved. Other researchers have also reported similar findings on the role of beliefs in engaging communities in the implementation of preventive measures [29,30].

The tendency to conceal symptoms from others and the perception of not being susceptible to COVID-19 among some members of the community may indicate a lack of adequate awareness about the comprehensive features of the disease in the early stages. This could also be related to an over-arousal of fear during health education or community mobilization campaigns. Hiding symptoms in order not to be taken to the isolation centers could have been easily addressed by convincing other family members instead of preventing communication when transferring the patient to the isolation centers.

In this study, various types of myths and misconceptions were identified as the main reasons for low COVID-19 vaccination rates and individual hesitancy. These include the belief that receiving the vaccine will cause infertility, blood clotting, or risk of death and the belief that the number of the beast (666) is associated with the vaccine. The sources of these beliefs are mainly social media platforms used by individuals and some groups/institutions, indicating that the need to monitor and correct such misinformation and engage with social media actors is demanding. Providing accurate information to counter these rumors and providing factual data about the disease and vaccine will significantly reduce the problem [31].

The connection between COVID-19 and the political situation of a country, as well as national elections, is another aspect indicating the need to monitor social media and actors in the field to avoid misinformation. Such types of misinformation were common before COVID-19 and need countering. Previous studies have reported the presence of misinformation during the COVID-19 pandemic. There have been recommendations to correct misinformation and build a presence of health professionals on social media, providing training to enable them [32]. Associating COVID-19 vaccination with infertility, while no evidence exists, may have unintended consequences for the future provision of an expanded program of immunization for children in the country and other diseases or similar emerging pandemics in the future, thus requiring attention.

In our study, some participants were not vaccinated due to a lack of trust in the vaccine itself; they questioned the availability of more than one type, and countries having their own while criticizing others. This was linked to news regarding the blood clotting effect of vaccines, whereas some mentioned that the time was not adequate to create a new vaccine. Handling such issues at global and national levels and working in collaboration, led by United Nation entities, to develop vaccines for newly emerging pandemics will help address this. The lack of trust might be due to inadequate awareness about the process of vaccine donation and how the Ministry of Health approved importing such vaccines. It may also indicate the need to provide adequate information and to set up awareness creation programs in the media or other alternatives. Studies among health professionals in Ethiopia [16,33] and a vaccine hesitancy scope review in Africa also found that a lack of trust in pharmaceutical industries or the vaccine’s effectiveness was a reason for vaccine hesitancy [34]. A.J. Calac et al. reported that healthcare professionals and all those working in the public health arena need to build and display trust in order to dispel the widespread dissemination and acceptance of false information in this technological age [35].

A fear of injection and the route of administration of the vaccine was another reason for not being vaccinated, indicating the need for more health education and awareness to foster a risk and benefit analysis at the individual level, using those who have already been vaccinated as champions. A previous quantitative study from the UK also reported that blood-injection-injury fears may explain approximately 10% of cases of COVID-19 vaccine hesitancy [36].

This study was conducted in the main cities of all regions, except Tigray, and could help to understand the various reasons for hesitancy towards COVID vaccination. We collected data using experienced qualitative research assistants by applying FGD and IDIs; therefore, the findings are credible. This study’s findings may explain the reasons behind the high level of COVID-19 vaccine hesitancy observed in national (64.4%) [4] and Addis Ababa (57%) [37] studies. However, this is qualitative research with the aim of gaining in-depth understanding; thus, it cannot be generalized to any region or at the national level. Future studies should consider a mixed-method design to estimate the types and magnitude of the reasons for the low uptake of COVID-19 vaccination.

## 5. Conclusions

Our findings showed that community engagement in the implementation of prevention efforts varied by city and did not sustain for an extended period during the pandemic. This may indicate the need for continuity of health communication at the national level and local community sensitization to effectively provide reliable information at the community level. The most common reason that participants gave for not being vaccinated was linked with misconceptions and rumors heard from social media; this indicates the need to recognize the role of social media and the actors in this business in future interventions during risk communication or community mobilization. Training healthcare providers on how to counter erroneous messages and enabling them to engage in social media are also worthwhile. Our findings also revealed that the community associated COVID-19 with evil and the number of the beast in all cities. Thus, working in close collaboration with religious institutions to address misconceptions associated with religion, and providing adequate training to healthcare providers regarding the vaccine, including the side effects and eligibility by cases, should be a priority.

Unproven or unapproved messages regarding traditional prevention measures or self-medication of herbal or other sources need to be monitored, and further investigation needs scientific evidence to avoid the negative effect that such traditional methods or natural products have on the implementation of scientific prevention measures or other elated side effects. Based on our findings, we recommend engaging and monitoring social media in such emergencies and addressing the issue by approving or disapproving any information on digital media. Further studies should be conducted to assess the types and magnitude of impacts from the myths and misconceptions on COVID-19 vaccination up-take.

## Figures and Tables

**Figure 1 vaccines-11-01511-f001:**
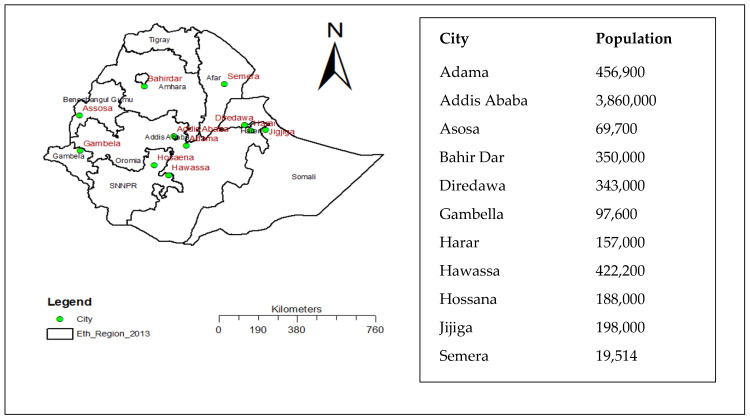
The distribution of study areas (cities).

**Table 1 vaccines-11-01511-t001:** List of main categories and sub-categories.

Main Category	Sub-Categories	Response Summaries
Early perception of COVID-19 and community reaction	Perceived as a supernatural punishment. Experienced fear and frustration. Feeling nothing and even some were happy.	COVID-19 as a punishment for sinners. Stop working and stayed at home. Relying on supernatural and linking with national election postponement.
Preventive measures and practices	Traditional methods: the only option in the beginning. Non-pharmaceutical interventions. Community engagement on preventive measures.	Using traditional herbal sources and traditional alcohols. Staying at home, physical distancing, hand hygiene and face mask use. More implemented in Addis Ababa. Varied with law enforcement. Not practiced at the time of data collection.
Vaccine awareness and sources of information	Vaccine awareness. Sources of information.	All are aware there is a vaccine. Some did not know whether it was free of not. Media, health professionals, volunteering group campaigns, family and friends were the sources.
Reasons for vaccine hesitancy	Misinformation and misconception. Lack of trust. Afraid of pain and side effects. Political bias. Lack of the right information for decision. Economic reasons. Misperception of absence of symptom among positive cases. Lack of awareness and negligence.	The vaccine is used to administer 666 (number of beast). It has microchips and the aim is to control individuals. The vaccine could lead to infertility. Vaccinated people would die in two years. Multiplicity of the types (why many?). Vaccines were developed in short time. Not well tested and wanted to test here. Afraid of the injection. Complain of pain among vaccinated. Will be sick after getting vaccinated. The government used to skip the election while no COVID-19. No proper information for pregnant and lactating mothers. Thinking vaccination will have pain and lead to absence from daily job. Believing the disease will not affect young and healthy. Not knowing the advantages and disadvantages of the vaccine, some did not want at all
Future intentions towards vaccination	Future intention.	Most want to be vaccinated in the future. Some do not want—claiming their reliance in the supernatural.
Suggestions on the way forward	Suggestions.	Continuous awareness creation. Working with religious leaders. Giving correct information in advance.

## Data Availability

The original raw data used in this study are available from the corresponding author and can be presented upon reasonable request.

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
