# Peer review of "Exploring Community Perceptions of COVID-19 and Vaccine Hesitancy in Selected Cities of Ethiopia: A Qualitative Study"

_vaccines, 2023, doi:10.3390/vaccines11101511_

Round 1
Reviewer 1 Report
While the manuscript demonstrated overall competence in terms of writing, there were some notable areas that could benefit from improvement. Specifically, there was a lack of sufficient transitions between paragraphs, important methodological details were missing, and it seemed that no theoretical framework was employed to guide the study and analysis.
Specific comments:
1. Please note that 'COVID-19' is 'Coronavirus Disease 2019'. Please define all abbreviations in the first instance of their use.
2. "... although almost 70% of vaccine hesitancy in Israel was due to safety problems (10)" - it is also relevant to mention in the introduction that in another study of anti-vaccine sentiments, the majority of the negative sentiment tweets towards COVID-19 vaccination were centered around criticisms regarding perceived coercive policies (citation: pubmed.ncbi.nlm.nih.gov/36146535). This could draw anger and ire among the public.
3. "... achieving a vaccination performance goal of at least 20% was not possible" - this was by the end of 2021 right?
4. "This study was conducted in August 2022" - over how long? And how should the period of study be considered? Is there 'fatigue' from the prolonged pandemic, or would people be less inclined to be vaccinated now that most countries have adopted a 'Freedom Day' posture? Does the choice of time period affect the findings of the study? If so, what is the impact?
4. What particular QUAL research tradition was used (i.e., phenomenology? case study)?
5. Please specify if this was an inductive or deductive content analysis.
6. What were the main codes? This should have been specified along with a frequency table. Most types of qualitative content analysis utilize frequency to identify which concepts may warrant further exploration.
7. I would have appreciated a more detailed description on how the themes/sub-themes were derived from the codes and whether they were developed by a second researcher working independently from each other.
8. What was the basis for ending up with 70 in-depth interview participants and 28 FGDs?
9. There are no shortage of published works (both quantitative and qualitative) regarding the problem of vaccine hesitancy in low-income, middle-income and high-income countries. In the present study, there were no themes generated that particularly showed the value of the study. Superstitious beliefs are well known in these communities (see: ncbi.nlm.nih.gov/pmc/articles/PMC9152622). Maybe you can revisit the analysis or initial theoretical framework and ask: how do these concerns/experiences influence vaccine acceptance and hesitancy? Are the findings unique or generalizable? In this way, you attempt to elevate your findings beyond what has already been redundantly explored in the existing literature.
10. There are many different sections, subsections and subsubsections and these should be more clearly and consistently numbered/labelled.
11. To avoid confusion with the reference order and citation, please consider using brackets “[]” for citation instead.
Minor edits required.
Author Response
Point-by-point response_ Reviewer 1
We are very thankful for the time and constructive comments from the reviewer. We have modified the contents of the background and method parts based on the comments provided by the reviewer and received proofreading from a native.
Here we have provided a point-by-point response to the comments. We have received and incorporated all the comments. All the comments provided are extremely helpful in enhancing our manuscript.
- Please note that 'COVID-19' is 'Coronavirus Disease 2019'. Please define all abbreviations in the first instance of their use.
Response: Thank you for the comment. Yes, we included acute respiratory syndrome coronavirus 2 (SARS-COV-2), the coronavirus disease of 2019 (COVID-19) has been added in the introduction part, in the first instance of its use.
- "... although almost 70% of vaccine hesitancy in Israel was due to safety problems (10)" - it is also relevant to mention in the introduction that in another study of anti-vaccine sentiments, the majority of the negative sentiment tweets towards COVID-19 vaccination were centered around criticisms regarding perceived coercive policies (citation: pubmed.ncbi.nlm.nih.gov/36146535). This could draw anger and ire among the public.
Response: This is very helpful, and we have incorporated the comment along with the reference. Thank you!
- "... achieving a vaccination performance goal of at least 20% was not possible" - this was by the end of 2021 right?
Response: Thank you for this comment. Now we have included the time ‘ till November 2021’.
- "This study was conducted in August 2022" - over how long? And how should the period of study be considered? Is there 'fatigue' from the prolonged pandemic, or would people be less inclined to be vaccinated now that most countries have adopted a 'Freedom Day' posture? Does the choice of time period affect the findings of the study? If so, what is the impact?
Response: Thank you for raising this concern. No specific criteria were used to choose the time and it took us almost one month to finish the data collection in all cities. We conducted the study when we secured the resources. As you know, we carried out the study in 11 cities, which was resource demanding. We tried to explore the perception of participants since the beginning of the pandemic so as they will depend on the present time during the response. Even though we did not consider the concerns raised by the reviewer, starting the interviews and focus group discussion with the exploration of their perception could help us to minimize the time effect.
What particular QUAL research tradition was used (i.e., phenomenology? case study)?
Response: This is also a valid concern. We applied qualitative content analysis and cited reference to specifying which specific qualitative content analysis we used or whose approach we followed under the methods part (study approach sub-section).
- Please specify if this was an inductive or deductive content analysis.
Response: Thank you for the comment. We applied an inductive approach, and we now include that under the data analysis sub-section based on the comment.
- What were the main codes? This should have been specified along with a frequency table. Most types of qualitative content analysis utilize frequency to identify which concepts may warrant further exploration.
Response: Yes, qualitative content analysis is known for the counting of the contents and presentation of frequency of those contents or codes. However, we used a non-frequency approach qualitative content analysis as we did not aim to see which codes were common as our aim was to explore all the possible reasons for the vaccine hesitancy. However, we have included a table which includes the list of categories and sub-categories along with summary responses based on this comment.
- I would have appreciated a more detailed description on how the themes/sub-themes were derived from the codes and whether they were developed by a second researcher working independently from each other.
Response: Now, we have included the steps of analysis and how the categories and sub-categories were derived inductively based on the meaning coming from the data.
- What was the basis for ending up with 70 in-depth interview participants and 28 FGDs?
Response: Thank you for the comment. We have included this under the data collection sub-section. As we conducted five in-depth interviews and two FGDs from a designated city, we achieved data saturation and therefore did not require any additional participants.
- There are no shortage of published works (both quantitative and qualitative) regarding the problem of vaccine hesitancy in low-income, middle-income and high-income countries. In the present study, there were no themes generated that particularly showed the value of the study. Superstitious beliefs are well known in these communities (see: ncbi.nlm.nih.gov/pmc/articles/PMC9152622). Maybe you can revisit the analysis or initial theoretical framework and ask: how do these concerns/experiences influence vaccine acceptance and hesitancy? Are the findings unique or generalizable? In this way, you attempt to elevate your findings beyond what has already been redundantly explored in the existing literature.
Response: Thank you for the suggestion and we incorporated the reference in the background section. In this study, we tried to include different urban settings of the country and explore the perception and vaccine hesitancy. Our aim was limited to understanding the perception and hesitancy and we may not generalize this qualitative finding to the whole. However, we believe that the findings have indicated multiple reasons for the vaccine hesitancy. We presented the draft findings to the Ministry of Health and other partners working in the area. Based on the feedback we received, they found it was useful for their interventions.
- There are many different sections, subsections and subsubsections and these should be more clearly and consistently numbered/labelled.
Response: Thanks. We numbered the main categories and the sub-categories based on the suggestion. In deeded this was very important suggestion to help the reader capture the flow.
- To avoid confusion with the reference order and citation, please consider using brackets “[]” for citation instead.
Response: Thank you. Yes, this is corrected now.
Reviewer 2 Report
This study lacks necessary statistical classification and analysis.
NA
Author Response
Point-by-point response_ Reviewer 2
We are very thankful for the time and comments from the reviewer. We have modified the contents of the background and method parts and received proofreading from a native.
One comment on the analysis part:
This study lacks necessary statistical classification and analysis.
Thank you for the concern. However, this is qualitative research and we do not have data to apply statistical analysis. We reported data using main and sub-categories along with thick description and direct quotes from the participants.
Reviewer 3 Report
The authors of this study have performed a careful and thoughtful qualitative analysis of 70 in-depth interviews and 28 focus groups in Ethiopia to explore community perceptions about the COVID-19 vaccine and misperceptions leading to vaccine hesitancy. Overall, the study is nicely done, but some clarifications are needed. The explanations of sample recruitment and justification are clear and convincing. The findings are important.
1. P. 2 lines 70-85: The first round of vaccine is Nov 2021 and the second is March 2021. This does not make sense and might be a mistake in the writeup.
2. P. 3 line 105 “NPI” abbreviation needs to be spelled out.
3. P. 4 line 156: were males and females in the same group? It is mentioned that efforts were made to avoid status differences of the observers and note takers (line 183), but what about among the participants?
4. P. 5 line 235: How were the categories and sub-categories arrived at and established for their reliability?
The findings seem to echo those of many studies from other countries about the themes of misinformation and the effects of misinformation on decision making and health behavior. In addition, the role of in-group community trust and political bias are well documented elsewhere. These are not weaknesses of the current study; indeed these can be framed as strengths in that they illustrate the pervasive nature of social media for spreading these themes and demonstrating that health communication theories appear to be applicable across many cultures. It is essential for the authors to bring these types of studies into their manuscript in order for their study to have its full impact. For example, there is an excellent piece that could get them started by Calac and Southwell about “ How Misinformation Research Can Mask Relationship Gaps that Undermine Public Health Response.” https://doi.org/10.1177/089011712110709 There are other cross-national pieces about how trust in institutions have contributed to misinformation problems and increasing disparities have contributed to the spread of distrust in government and elites.
Only minor issues such as a missing article here and there, for example on line 136 "a" is missing between "using" and "qualitative"
Author Response
Point by point response_ Reviewer 3
We are very thankful for the time and constructive comments from the reviewer. We have modified the contents of the background and method parts based on the comments provided by the reviewer and received proofreading from a native.
Here we have provided a point-by-point response to the comments. We have received and incorporated all the comments. All the comments provided are extremely helpful in enhancing our manuscript.
- P. 2 lines 70-85: The first round of vaccine is Nov 2021 and the second is March 2021. This does not make sense and might be a mistake in the writeup.
Response: We are very thankful to this reviewer for picking up this error during typing but all the co-authors did not notice. Now, corrected.
- P. 3 line 105 “NPI” abbreviation needs to be spelled out.
Response: Thanks. Sure, this has been spelled out in line 39 or third line of the introduction . It is well acknowledged that non-pharmaceutical interventions (NPIs) have…
- P. 4 line 156: were males and females in the same group? It is mentioned that efforts were made to avoid status differences of the observers and note takers (line 183), but what about among the participants?
Response: Thanks for this comment. We added a sentence about the process of conducting separate FGDs for males and females.
To be culturally sensitive separate FGDs were conducted for males and females.
- P. 5 line 235: How were the categories and sub-categories arrived at and established for their reliability?
Response: Now, we have included the steps of analysis and how the categories and sub-categories were derived inductively based on the meaning coming from the data in the data analysis sub-section.
The findings seem to echo those of many studies from other countries about the themes of misinformation and the effects of misinformation on decision making and health behavior. In addition, the role of in-group community trust and political bias are well documented elsewhere. These are not weaknesses of the current study; indeed these can be framed as strengths in that they illustrate the pervasive nature of social media for spreading these themes and demonstrating that health communication theories appear to be applicable across many cultures. It is essential for the authors to bring these types of studies into their manuscript in order for their study to have its full impact. For example, there is an excellent piece that could get them started by Calac and Southwell about “ How Misinformation Research Can Mask Relationship Gaps that Undermine Public Health Response.” https://doi.org/10.1177/089011712110709 There are other cross-national pieces about how trust in institutions have contributed to misinformation problems and increasing disparities have contributed to the spread of distrust in government and elites.
Response: Thank you for the feedback and suggestion. We have incorporated this in the discussion section and along with the citation.
Round 2
Reviewer 1 Report
There are still several copyediting and content issues.
1. Please change "coronavirus disease of 2019 (COVID-19)" to "Coronavirus Disease 2019 (COVID-19)".
2. Please change "... liable to generate public frustration and anger" to "... which can generate public frustration and anger".
3. What is the theoretical framework and initial hypothesis for the qualitative study? Sadly the state of theory in this paper is still rather lacking.
4. "... non-frequency approach qualitative content analysis as we did not aim to see which codes were common as our aim was to explore all the possible reasons for the vaccine hesitancy" - I am not sure I agree with this. This is still a qualitative analysis even if you do present the frequencies.
5. Please change "non-pharmaceutical preventive measures" to "non-pharmacologic preventive measures".
6. Please standardize the citation style, the journal name should be italicized and not the title of the paper.
Moderate edits required.
Author Response
Point-by-point response_ Reviewer 1
Thank you very much for taking the time to review this manuscript and for your continuous constructive comments. Please find the detailed responses below.
- Please change "coronavirus disease of 2019 (COVID-19)" to "Coronavirus Disease 2019 (COVID-19)".
Response: Corrected. Thank you.
- Please change "... liable to generate public frustration and anger" to "... which can generate public frustration and anger".
Response: Corrected. Thank you.
- What is the theoretical framework and initial hypothesis for the qualitative study? Sadly the state of theory in this paper is still rather lacking.
Response: Thank you for the concern. As this is qualitative research, we did not have any theoretical framework and initial hypothesis.
- "... non-frequency approach qualitative content analysis as we did not aim to see which codes were common as our aim was to explore all the possible reasons for the vaccine hesitancy" - I am not sure I agree with this. This is still a qualitative analysis even if you do present the frequencies.
Response: Yes, this is qualitative research, and our aim was to better understand all the possible reasons for the vaccine hesitancy. We did not aim to count how many times each code appeared, or which ones are most common. Therefore, we do not have such an output to present here.
- Please change "non-pharmaceutical preventive measures" to "non-pharmacologic preventive measures".
Response: Thank you for your concern. Non-pharmaceutical preventive measures here is to refer the WHOs NPIs including hand washing/using sanitizers, face mask use and physical distancing. As readers are more familiar with terms would like to keep it. However, based on this concern, we modified the contents to make it clear. The participants listed various non-pharmaceutical measures practiced for COVID-19 prevention.
- Please standardize the citation style, the journal name should be italicized and not the title of the paper.
Response: Thank you for this important suggestion. The formatting was done by the journal, and we will inform them to correct it. Much appreciated!
Comments on the Quality of English Language: Moderate edits required.
Response: Thank you for the suggestion. This manuscript has undergone final proofreading by a native English speaker.
Reviewer 3 Report
I think everything is fine now although I would prefer to see more discussion of theory and previous cross-national studies that demonstrate the pervasiveness and validity of the findings the authors have demonstrated so beautifully in this set of findings, because I think their work would have more impact if they did so.
Author Response
Response_ Reviewer 3
Thank you very much for taking the time to review this manuscript and for your continuous constructive comments. Please find our response below.
I think everything is fine now although I would prefer to see more discussion of theory and previous cross-national studies that demonstrate the pervasiveness and validity of the findings the authors have demonstrated so beautifully in this set of findings, because I think their work would have more impact if they did so.
Response: Thank you for the suggestion. We have added one sentence which indicates quantitative findings showing level of vaccine hesitancy in the country. This study’s findings may explain the reasons behind the high level of COVID-19 vaccine hesitancy observed in national (64.4%) [4] and Addis Ababa (57%) [39] studies.